Li *et al. Genome Biology* (2022) 23:51

**RESEARCH**

# A large-scale genome and transcriptome sequencing analysis reveals the mutation landscapes induced by high-activity adenine base editors in plants

Shaofang Li[1*†], Lang Liu[1,2,3†], Wenxian Sun[3], Xueping Zhou[1,4] and Huanbin Zhou[1,2*]

*Correspondence:
shaofangli2021@hotmail.com; zhouhuanbin@caas.cn
†Shaofang Li and Lang Liu contributed equally to this work.
[1] State Key Laboratory for Biology of Plant Diseases and Insect Pests, Institute of Plant Protection, Chinese Academy of Agricultural Sciences, Beijing 100193, China
[2] Scientific Observing and Experimental Station of Crop Pests in Guilin, Ministry of Agriculture and Rural Affairs, Guilin 541399, China
Full list of author information is available at the end of the article

## Abstract

**Background:** The high-activity adenine base editors (ABEs), engineered with the recently-developed tRNA adenosine deaminases (TadA8e and TadA9), show robust base editing activity but raise concerns about off-target effects.

**Results:** In this study, we perform a comprehensive evaluation of ABE8e- and ABE9-induced DNA and RNA mutations in Oryza sativa. Whole-genome sequencing analysis of plants transformed with four ABEs, including SpCas9n-TadA8e, SpCas9n-TadA9, SpCas9n-NG-TadA8e, and SpCas9n-NG-TadA9, reveal that ABEs harboring TadA9 lead to a higher number of off-target A-to-G (A>G) single-nucleotide variants (SNVs), and that those harboring CRISPR/SpCas9-NG lead to a higher total number of off-target SNVs in the rice genome. An analysis of the T-DNAs carrying the ABEs indicates that the on-target mutations could be introduced before and/or after T-DNA integration into plant genomes, with more off-target A>G SNVs forming after the ABEs had integrated into the genome. Furthermore, we detect off-target A>G RNA mutations in plants with high expression of ABEs but not in plants with low expression of ABEs. The off-target A>G RNA mutations tend to cluster, while off-target A>G DNA mutations rarely clustered.

**Conclusion:** Our findings that Cas proteins, TadA variants, temporal expression of ABEs, and expression levels of ABEs contribute to ABE specificity in rice provide insight into the specificity of ABEs and suggest alternative ways to increase ABE specificity besides engineering TadA variants.

**Keywords:** Adenine base editor, TadA variants, Single nucleotide variant, T-DNA insertion, Off-target, *Oryza sativa* L

## Background

Single-nucleotide variants (SNVs), a universal feature of plant, animal, and human genomes, have been widely identified in association with agronomic traits and human diseases [1–3]. Various clustered regularly interspaced short palindromic repeats

(CRISPR)/CRISPR-associated protein (Cas)-mediated base editing tools (e.g., ABEs and cytosine base editors), which efficiently produce desired point mutations in genomic DNA without causing double-stranded DNA breaks [4], have been used widely in laboratory research, crop and animal breeding, as well as human gene therapy [5–7]. Since the mutation of G•C base pairs to A•T base pairs is the primary form of de novo mutations [8], ABEs that catalyze the conversion of A•T base pairs to G•C base pairs have great potential to correct human pathogenic point mutations [9]. However, potential DNA and RNA off-target mutations remain a serious concern and threaten to limit the application of ABEs.

The pioneer ABE7s, which are composed of a tRNA adenosine deaminase (TadA7.10) and CRISPR/Cas systems, perform remarkably clean and efficient A•T to G•C conversions in the genomes of a variety of species, including human, mouse, and rice, without inducing obvious genome-wide off-target DNA mutations [10–14]. However, the editing efficiency of ABE7s varies in a locus-dependent manner [11, 13]. Subsequently, high-activity ABEs, such as those containing TadA8.17, TadA8.20, TadA8e, and TadA9, have been developed, engineered with various PAM-flexible Cas variants and tested in different organisms [15–18], but a whole-genome assessment of the off-target DNA mutations induced by TadA8e and TadA9 has not yet been investigated.

The tRNA adenosine deaminase TadA, a key component of ABEs, induces site-specific inosine formation on RNAs [19]. Recently, it was reported that TadAs, ABE7s, and ABE8es induced a significantly higher number or higher mutation ratio of RNA A-to-G (A>G) SNVs when compared to Cas proteins or GFP [9, 20, 21] and that ABE8.17 and ABE8.20 induced very low levels of adenosine deamination in mRNAs if ABEs were delivered as messenger RNAs in mammalian cells [17]. Thus, several labs have developed improved TadA variants with reduced RNA activity [20, 22]. However, RNA A>G mutations induced by ABEs are complicated due to the large genomes in the heterogeneous mammalian cells as well as the conversion of adenosines into inosines mediated by endogenous adenosine deaminase RNA specific (ADAR) family. In addition, ABE-induced RNA mutations have never been reported in plant yet.

The relatively small genome (~ 0.4 Gb) of self-pollinated rice and the absence of endogenous ADAR family make rice an ideal model organism to examine the DNA and RNA specificity of gene editing tools. Here, we investigated the off-target DNA and RNA mutations induced by ABE8es and ABE9s in rice through whole-genome sequencing (WGS) and transcriptome sequencing.

## Results

### ABEs induced sgRNA-independent heterozygous DNA mutations

To assess the off-target activity effects of high-activity tRNA adenosine deaminases (TadA8e and TadA9), we chose four ABEs that are composed of different variants of tRNA adenosine deaminase and CRISPR/Cas systems with different PAM compatibility: rBE46b (SpCas9n-TadA8e), rBE49b (SpCas9n-TadA9), rBE50 (SpCas9n-NG-TadA8e), and rBE53 (SpCas9n-NG-TadA9) (Fig. 1a). For each ABE, three constructs with one or two sgRNAs and one construct without sgRNAs were generated. After *Agrobacterium tumefaciens*-mediated transformation, we obtained three independent transgenic plants for each construct except 46bM and 49bM, which had three plants from

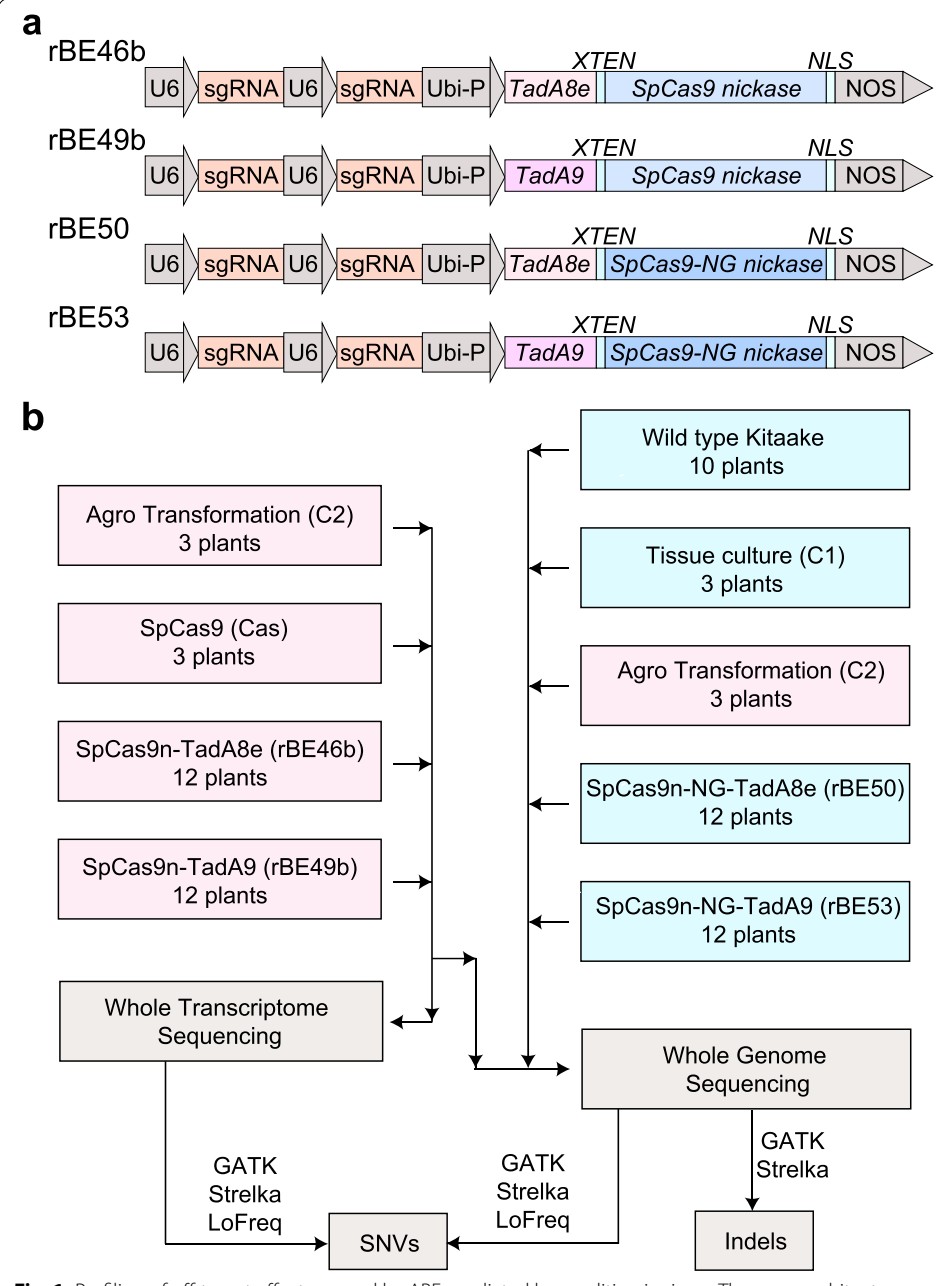

**Fig. 1** Profiling of off-target effects caused by ABE-mediated base editing in rice. **a** The gene architecture of four base editors: rBE46b, rBE49b, rBE50, and rBE53. Ubi-P, maize ubiquitin 1 promoter, *NLS*, nuclear localization sequence; NOS, nopaline synthase terminator. **b** Diagram of the experimental design. For plants in pink rectangles, both genomes and transcriptomes were sequenced. For plants in blue rectangles, only genomes were sequenced

two independent transformation events (Additional file 1: Table S1). We examined the on-target mutations in the 36 plants carrying the ABE plus sgRNA(s) through Sanger sequencing and identified the desired mutations in 35 plants (Additional file 2: Fig. S1 and S2). To assess the effects of tissue culture and *Agrobacterium* infection, three independently regenerated plants subjected to tissue culture and six independently

regenerated plants subjected to *Agrobacterium* infection without vectors were selected for WGS. We also sequenced 10 wild-type rice variety Kitaake plants to filter out background mutations (Fig. 1b). To ensure high confidence in base calling, we sequenced all 71 plants at an average coverage of 41× (Additional file 1: Table S2). SNVs in each plant were identified using three independent variant-calling software systems: GATK, Strelka2, and Lofreq [14, 23–25]. Small insertions or deletions (indels) were called independently by GATK and Strelka2. SNVs identified by all three methods and indels identified by two methods were kept for later analysis (Fig. 1b). All the SNVs and indels identified in the 10 Kitaake plants were considered background mutations and removed from the analysis. The sgRNA-guided on-target and off-target loci were located by Criflash [26] (Additional file 1: Table S3). Consistent with Sanger sequencing results, A>G on-target mutations were observed in 35 out of 36 plants (Additional file 2: Fig. S3) and removed in the following off-target analysis. No mutations were detected at 33 predicted sgRNA-guided off-target sites with 2–3 nt mismatches. To further examine sgRNA-dependent DNA mutations, we applied very loose standards to locate sgRNA-guided off-target sites. Only 5 SNVs were detected at 33,558 sgRNA-guided off-target sites located using Criflash with maximal 6 nt mismatches. It was reported that 2 nt mismatches within 12 nt seed sequence at the 3' end of the spacer would abolish the activity of SpCas9 [27]. Only 35 SNVs were detected at 366,136 sgRNA-guided off-target sites by searching the 12 nt seed sequence with maximal 2 nt mismatches. These facts demonstrate the ABEs with highly specific sgRNAs rarely induce sgRNA-dependent off-target mutations under our conditions. For plants that had undergone tissue culture (control group 1: C1) and *Agrobacterium* infection (control group 2: C2), we identified around 200–400 SNVs and around 250–350 indels from each plant (Additional file 2: Fig. S4a, b). For plants carrying an ABE, we identified around 200–800 SNVs and 200–500 indels (Additional file 2: Fig. S4a, b). Six types of SNVs were identified separately in control plants and in those carrying ABEs. We discovered that A>G/T>C SNVs constituted a higher proportion in plants with ABEs (Additional file 2: Fig. S4c, d). For simplicity, we referred to the number of A>G SNVs as the total number of A>G and T>C SNVs, and we referred to the percentage of A>G SNVs as the percentage of the total number of A>G and T>C SNVs versus the total number of all six types of SNVs throughout the manuscript. Consistently, the number and percentage of A>G SNVs in plants with ABEs were higher when compared to both control groups, indicating that ABEs induce the genomic mutations of the A•T base pairs to G•C base pairs (Additional file 2: Fig. S4c-f).

A few homozygous SNVs and indels were detected in all sequenced plants (Additional file 2: Fig. S5a, b). We counted the number of plants with the same mutation sites and found that the homozygous mutations tended to be present in more than one plant, while the heterozygous mutations tended to be present in a single plant (Additional file 2: Fig. S5c, d). These homozygous mutations could be the remaining background mutations or mutations induced by tissue culture, *Agrobacterium* infection, or ABEs. The induced mutations in the two alleles are two independent events following binomial distribution, so the probability of the homozygous mutations is $p^2$, the probability of being wild type (WT) is $(1-p)^2$, and the probability of the heterozygous mutations is $2 * p * (1-p)$, assuming that the induced mutation ratio for each allele was p and the ratio of the WT allele was 1-p. A binomial test for all loci of homozygous SNVs or indels

revealed that these loci did not follow a binomial distribution (Additional file 2: Fig. S5e and Additional file 1: Tables S4 and S5), indicating that these homozygous mutations remain background SNVs and indels. These data suggests that ABEs induce sgRNA-independent heterozygous DNA mutations.

### Genome-wide analysis of ABE-induced single-nucleotide mutations

After background homozygous mutations were removed, we recalculated the number of SNVs and indels in the plants (Additional file 2: Fig. S6a, b and Additional file 1: Table S6). We did not observe any significant differences of SNVs or indels induced by tissue culture or *Agrobacterium* infection (Fig. 2a, b). Therefore, we used the plants that

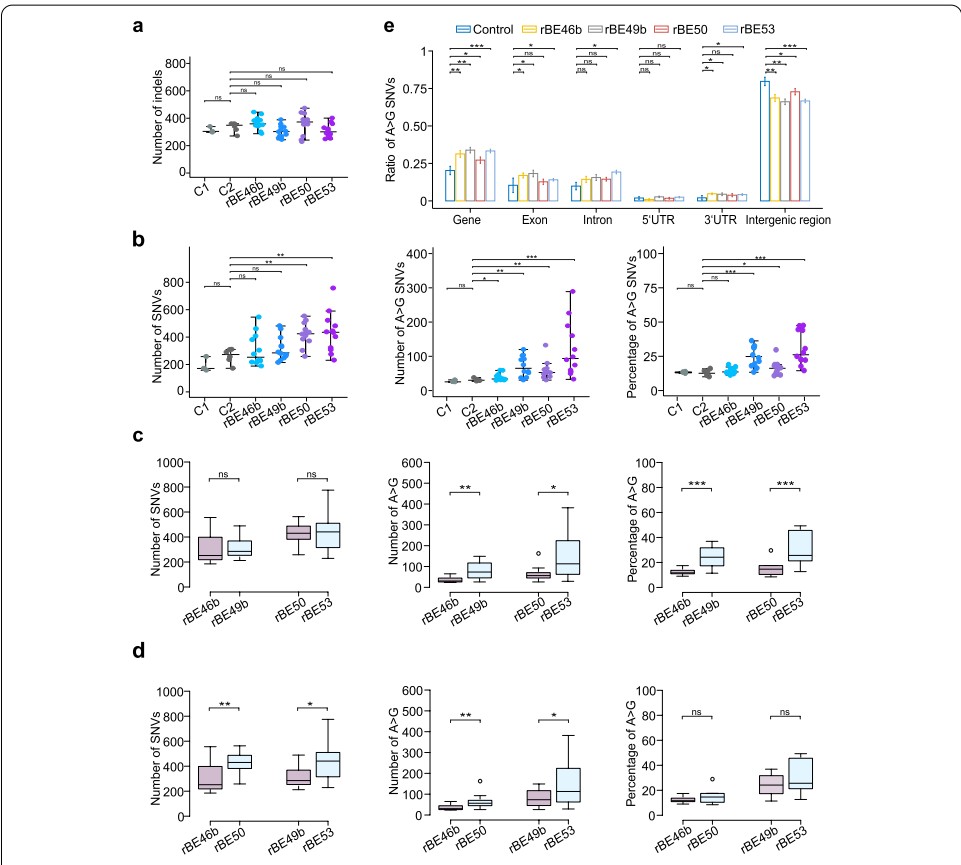

**Fig. 2** Characterization of ABE-induced genomic mutations. **a**, **b** Number of indels, SNVs, and A>G SNVs, and percentage of A>G SNVs identified for plants that had undergone tissue culture (C1) or *Agrobacterium* infection (C2) and plants harboring SpCas9n-TadA8e (rBE46b), SpCas9n-TadA9 (rBE49b), SpCas9n-NG-TadA8e (rBE50), and SpCas9n-NG-TadA9 (rBE53). In each plot, each dot represents the number of indels, SNVs, and A>G SNVs, and the percentage of A>G SNVs from an individual plant; each middle line represents the median value; and each upper line and lower line represent the standard errors. **c** Number of SNVs and A>G SNVs, and percentage of A>G SNVs were compared for ABE-edited plants harboring TadA8e or TadA9: rBE46b versus rBE49b, and rBE50 versus rBE53. **d** Number of SNVs and A>G SNVs, and percentage of A>G SNVs were compared for ABE-edited plants harboring SpCas9n or SpCas9n-NG: rBE46b versus rBE50, and rBE49b versus rBE53. **e** Percentage of A>G SNVs at given regions for plants in control groups or carrying one of the four ABEs. Each bar represents the mean value, and each error bar represents the standard error. (ns) denotes *p*-value > 0.1, (*) denotes *p*-value < 0.1, (**) denotes *p*-value < 0.01, and (***) denotes *p*-value < 0.001 (one-tailed Wilcoxon test)

had been infected with *Agrobacterium* as the control group in the following analysis. Consistent with the finding that ABEs do not cause double-strand DNA breaks, plants with ABEs did not show a higher number of indels (Fig. 2a). The number of total SNVs and A>G SNVs was significantly higher in plants harboring rBE50 and rBE53 than in the control groups, while the number and the percentage of A>G SNVs were significantly higher in plants harboring rBE49b and rBE53 than in plants in control groups (Fig. 2b). We did not observe a significantly higher number of SNVs or a higher percentage of A>G SNVs in plants harboring rBE46b (Fig. 2b).

We next examined whether Cas proteins or TadA variants play distinct roles in inducing off-target DNA mutations by comparing plants harboring rBE46b with those harboring rBE49b as well as rBE50 to rBE53 to characterize TadA8e and TadA9, and compared plants with rBE46b to rBE50 and rBE49b to rBE53 to characterize the role of SpCas9n and SpCas9n-NG in off-target effects. Although there was no significant difference between TadA8e and TadA9 when the total number of SNVs was considered, plants harboring TadA9 had a higher number and a higher percentage of A>G SNVs (Fig. 2c), indicating that TadA9-based ABEs lead to a higher number of A>G SNVs. Plants harboring SpCas9n-NG had a higher number of SNVs as well as a higher number of A>G SNVs, but not a higher percentage of A>G SNVs (Fig. 2d), indicating that SpCas9n-NG-based ABEs lead to a higher number of SNVs.

We classified all SNVs into six types and calculated the percentage of each type of SNV versus the total number of SNVs. We observed a higher percentage of C>A/G>T SNVs in plants harboring TadA8e (Additional file 2: Fig. S7). We further mapped all SNVs and A>G SNVs to different genic and intergenic regions and calculated the ratio of SNVs in given regions versus in the whole genome. As a result, the number of A>G SNVs and the total number of SNVs were higher at all genic and intergenic regions in plants for all four types of ABEs, while A>G SNVs were enriched in genic regions and depleted in intergenic regions (Fig. 2e and Additional file 2: Fig. S8). In addition, we mapped total SNVs as well as A>G SNVs to the 12 rice chromosomes and established that they were distributed throughout the rice genome (Additional file 2: Fig. S9).

### T-DNA insertion influences the single-nucleotide mutations

We detected genome-wide off-target SNVs induced by tissue culture from three plants, those induced by *Agrobacterium* infection without vectors in six plants, and those in 48 plants transformed by *Agrobacterium* infection with ABEs. We compared SNVs from the individual plants to those identified in all other plants to examine the overlapping SNVs. Among 1596 comparisons, we found none of the common SNVs in 1567 comparisons, and 1–7 overlapping SNVs in 27 comparisons (Additional file 1: Table S7), which indicates the randomness of off-target DNA mutations induced by tissue culture, *Agrobacterium* infection, and ABEs. In addition, we detected 147 overlapping SNVs in the comparison of lines 46bM_s2 and 46bM_s3, and 85 overlapping SNVs in the comparison of lines 49bM_s2 and 49bM_s3. Notably, 46bM_s2 and 46bM_s3 as well as 49bM_s2 and 49bM_s3 are plants regenerated from the same resistant calli (Additional file 1: Table S1). The T-DNA insertion sites in the genomes of three plants transformed with 46bM and three plants transformed with 49bM were located by T-LOC (Li et al. in preparation). We determined that lines 46bM_s2 and 46bM_s3 were derived from the

same T-DNA integration event, whereas line 46bM_s1 was from a different T-DNA integration event (Fig. 3a). Similarly, lines 49bM_s2 and 49bM_s3, but not line 49bM_s1, harbored the same T-DNA insertion site (Fig. 3a). Surprisingly, plants carrying the same T-DNA insertion event did not always have the same sgRNA-guided on-target mutations (Additional file 2: Fig. S3). To validate this phenomenon, we also sequenced line

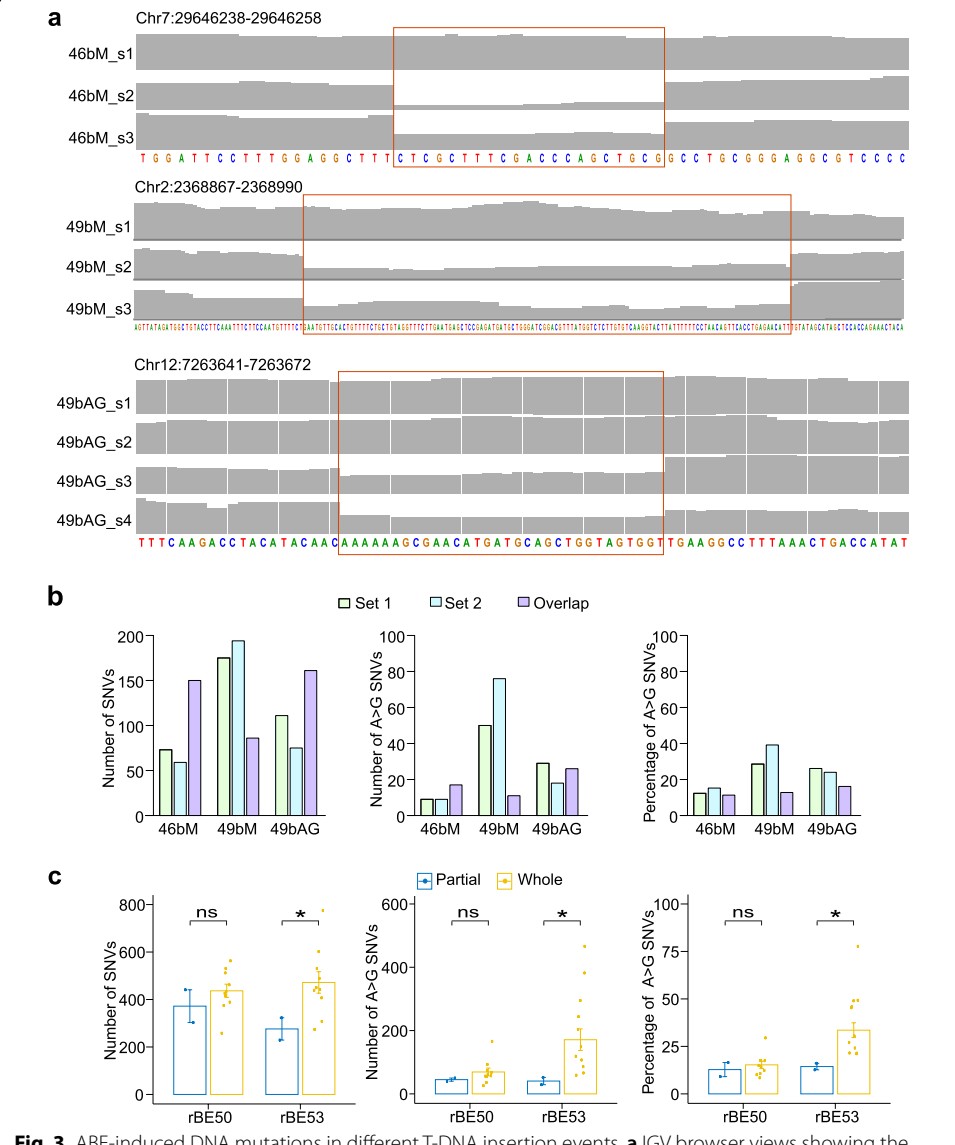

**Fig. 3** ABE-induced DNA mutations in different T-DNA insertion events. **a** IGV browser views showing the read coverages at T-DNA insertion sites. Lines 46bM_s2 and 46bM_s3, 49bM_s2 and 49bM_s3, and 49bAG_s3 and 49bAG_s4 were germinated from the same calli. Regions in red rectangles are the T-DNA insertion sites. **b** Number of SNVs and A>G SNVs, and percentage of A>G SNVs. Set 1 represents the unique SNVs only in 46bM_s2, 49bM_s2, and 49bAG_s3. Set 2 represents the unique SNVs only in 46bM_s3, 49bM_s3, and 49bAG_s4. Overlap represents the overlapping SNVs in 46bM_s2 and 46bM_s3, 49bM_s2 and 49bM_s3, and 49bAG_s3 and 49bAG_s4. **c** Number of SNVs and A>G SNVs, and percentage of A>G SNVs in plants with partial or whole T-DNA insertions of rBE50 or rBE53. Each bar represents the mean value, each error bar represents the standard error, and each dot represents the number of SNVs, the number of A>G SNVs, and percentage of A>G SNVs of each plant. (ns) denotes *p*-value > 0.1, (*) denotes *p*-value < 0.1 (one-tailed Wilcoxon test)

49bAG_s4, which was regenerated from the same resistant callus as line 49bAG_s3. We established that lines 49bAG_s3 and 49bAG_s4 had the same T-DNA insertion site, which differed from that of lines 49bAG_s1 and 49bAG_s2 (Fig. 3a), and that lines 49bAG_s3 and 49bAG_s4 had different on-target editing events (Additional file 2: Fig. S10a). We further characterized the off-target SNVs in these plants and found that different plants with the same T-DNA insertion had both unique SNVs and common SNVs (Fig. 3b and Additional file 2: Fig. S10b). We defined three sequential stages in *Agrobacterium*-transformed callus: stage 1, the period after the T-DNA plasmid has entered the callus cell and before it has integrated into the genome; stage 2, the period after the T-DNA has integrated into the genome and before the callus cell has divided; and stage 3, the period after the callus cell has divided. Since the off-target mutation happens randomly, the unique SNVs should occur at stage 3, while common SNVs should occur at both stage 1 and stage 2. A higher percentage of A>G SNVs was observed among the unique SNVs when compared to common SNVs in plants transformed with 49bM and 49bAG (Fig. 3b), indicating that ABEs integrated into the reference genome are more prone to cause A>G SNVs.

We next examined the integrity of T-DNA regions containing both a complete left border (LB) and right border (RB) and identified four plants with a partial T-DNA insertion characterized by the missing TadA8e, TadA9, or SpCas9n-NG fragment (Additional file 2: Fig. S11a). However, desired on-target mutations were detected in three out of four plants (Additional file 2: Fig. S3), suggesting that sgRNA-dependent on-target A>G editing could occur before T-DNA integration into the rice genome. We further checked the off-target SNVs between plants with or without complete T-DNA insertion and found that plants with a complete T-DNA insertion had a higher number of total SNVs, a higher number of A>G SNVs, and a higher percentage of A>G SNVs when compared to those with partial T-DNA insertion (Fig. 3c).

It was known that T-DNAs can be integrated in rice genome in more than one copy [28], so we divided the plants into two groups based on whether one copy or multiple copies of T-DNAs were integrated. We examined the number of total SNVs, the number of A>G SNVs, and the percentage of A>G SNVs in plants with rBE46b, rBE49b, rBE50 and rBE53 separately and did not observe a consistent influence of the copy number of T-DNA insertion (Additional file 2: Fig. S12).

### ABEs induce transcriptome-wide A>G RNA mutations

To examine whether ABEs induce RNA off-target mutations, we profiled the transcriptomes of three plants subjected to *Agrobacterium* infection without vectors, three transformed plants carrying functional SpCas9 only, three plants carrying rBE46b without sgRNAs, nine plants carrying rBE46b with one or two sgRNAs, three plants carrying rBE49b without sgRNAs, and nine plants carrying rBE49b with one or two sgRNAs (Fig. 1b). SNVs were called independently by GATK, Strelka2, and Lofreq from each transcriptome and the corresponding genome data. We kept SNVs called by all three methods in transcriptome data but not in genome sequencing data. In addition, SNVs detected from plants in the *Agrobacterium* infection group were removed as background mutations. Overall, the number of SNVs, the number of A>G SNVs, and the percentage of A>G SNVs were not significantly higher in plants harboring rBE46b and rBE49b

than in those harboring SpCas9 nuclease (Additional file 2: Fig. S13a and Additional file 1: Table S9); however, A>G SNVs constituted a higher proportion in plants harboring rBE46b and rBE49b than in plants harboring SpCas9 nuclease only (Additional file 2: Fig. S13b). When SNVs were counted separately for each plant, we found that transcriptomes R49AG_s2 and R49AG_s3 had more than 100 A>G SNVs and that A>G SNVs were barely detected in plants harboring SpCas9 only (Fig. 4a). In contrast to the randomness of DNA off-target SNVs, we found that the ratio of ABE-induced RNA off-target SNVs in transcriptomes R49AG_s2 and R49AG_s3 A>G correlated with each other (Fig. 4b) and that SNVs loci were commonly shared in the eight transcriptomes with more than 5 detected SNVs (Additional file 2: Fig. S14), indicating that ABEs might have preferred RNA editing sequence content. As expected, we identified a conserved YAN-enriched (Y = T, C and N = A, T, C, G) motif at ABE-edited RNA loci (Fig. 4c). We combined the SNV loci detected in all transcriptomes as ABE-targeting RNA loci, computed the A>G editing ratio in each transcriptome with sufficient read coverage (read number higher than 10), and performed a Wilcoxon test that compared the A>G editing ratio of each plant containing ABEs versus the A>G editing ratio in three plants that contained SpCas9. Although the number of ABE-targeted RNA loci with sufficient reads were comparable in all sequenced transcriptomes (Additional file 2: Fig. S13c), transcriptomes from eight plants harboring rBE46b and rBE49b, including R46AG_s1, R46AG_s3, R46GG_s1, R49AG_s2, R49AG_s3, R49bg_s1, R49bg_s2, and R49bg_s3, had significantly higher A>G editing ratios (Fig. 4d). Since these eight plants also had detectable numbers of A>G RNA SNVs, we concluded that ABEs (rBE46b and rBE49b) induced RNA editing in these eight plants but not in the remaining 16 plants. We examined the ABE-induced DNA off-target mutations, but found no differences between the plants with RNA mutations and those without RNA mutations (Additional file 2: Fig. S13e). When the reads per million (RPM) value of ABEs (SpCas9n/SpCas9n-NG and TadA8e/TadA9) was calculated, we found that the transcript levels of ABEs were significantly higher in the eight plants with RNA mutations than in the 16 plants without RNA mutations (Fig. 4e). Given the high concordance between ABE transcript abundance and the A>G editing ratio, we wondered whether RNA A>G editing would cease after the T-DNA insertion segregated out in the next generation. Two transgenic and two transgene-free plants were selected in the $T_1$ population of line 49AG_s2 and subjected to transcriptome analysis. As expected, A>G RNA editing was eliminated in the two $T_1$ plants that lacked the ABE transgene but remained active in the two plants with transgenes (Fig. 4f and Additional file 2: Fig. S13d).

### ABEs induce clustered off-target editing

Given that ABEs lead to multiple A>G editing events at the sgRNA-dependent on-target window, we wondered whether they function the same way at the A>G off-target editing loci. We examined the A>G mutations located within the 5′ and 3′ 30-bp flanking region of every ABE-induced A>G off-target locus in the transcriptome data. After counting A>G SNVs for which the A>G conversion rate was higher than 0.05 and also counting A sites in cases where the read coverage was higher than 10, we determined the ratios of A>G SNVs at every flanking position. In eight transcriptomes with RNA off-target editing, A>G SNVs were consistently distributed in the flanking regions (Fig. 5a). We refer

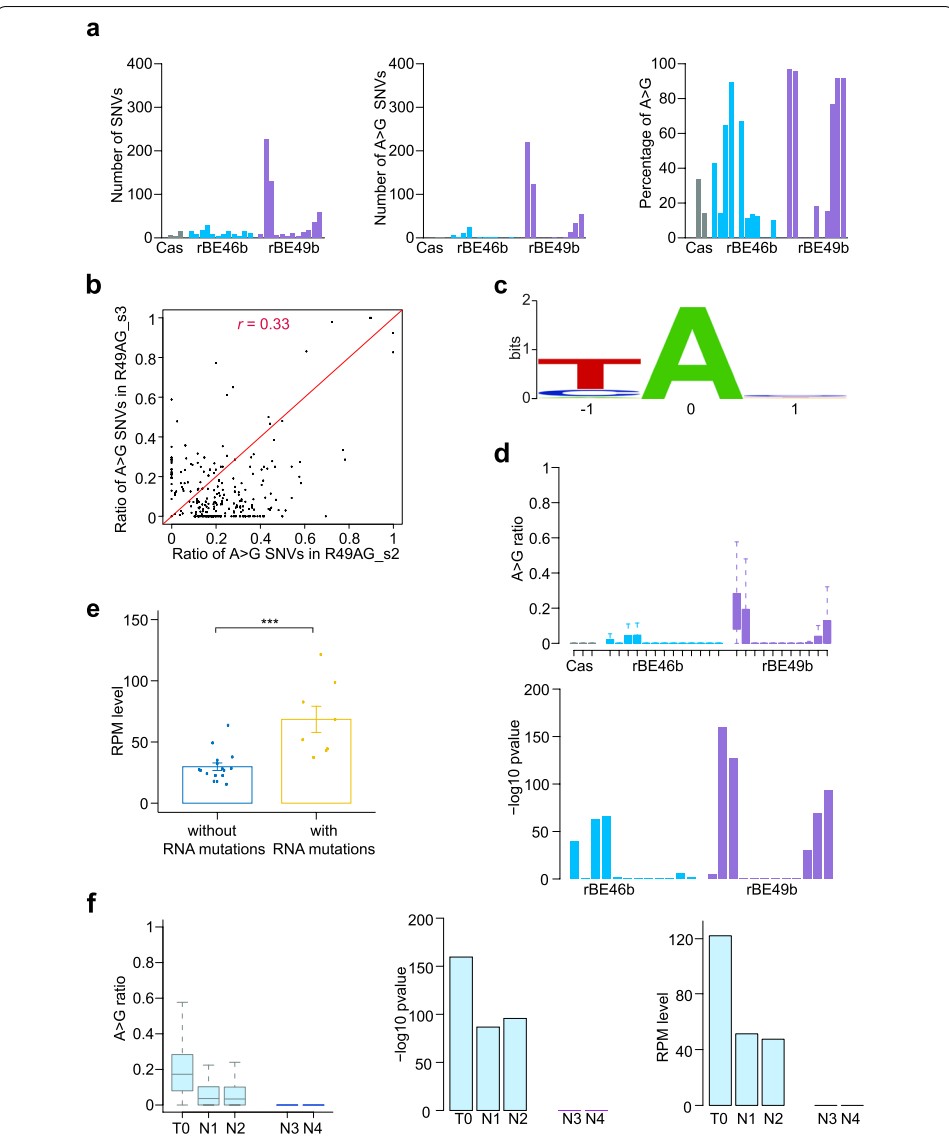

**Fig. 4** Transcriptome-wide ABE-induced off-target mutations. **a** Number of SNVs and A>G SNVs, and percentage of A>G SNVs in plants harboring SpCas9 (Cas), SpCas9n-TadA8e (rBE46b), and SpCas9n-TadA9 (rBE49b). **b** Ratios of A>G mutations were calculated for A>G SNV loci detected in lines R49bAG_s2 and R49bAG_s3 and shown in the scatterplot. The Pearson correlation coefficient (*r*) was also calculated, and the red line is the diagonal line. **c** A sequence logo derived from edited adenines from all RNA-seq data. Bits account for how much each column is conserved and how much the nucleotide frequencies obtained in the profile differ from those that would have been obtained by aligning oligonucleotides chosen at random. **d** Boxplot showing ratios of A>G mutations at all RNA A>G SNV loci for plants harboring SpCas9, rBE46b, and rBE49b. A Wilcoxon test was conducted between every plant harboring ABEs versus plants harboring Cas only, and the -log10 *p*-value is shown. **e** Bar plot showing the average RPM values of ABEs for plants without RNA mutations and plants with RNA mutations. Each bar represents the mean value, each error bar represents the standard error, and each dot represents the ABE RPM value of each plant. (\*\*\*) denotes *p*-value < 0.001 (one-tailed Wilcoxon test). **f** Ratios of A>G mutations of all A>G RNA SNV loci were calculated for one 49bAG_s2 $T_0$ plant and four 49bAG_s2 $T_1$ plants (left). -log10 *p*-value of Wilcoxon test on A>G ratios between five 49bAG_s2 plants versus plants harboring SpCas9 (middle). RPMs of ABEs are shown in the bar plot (right). N1 and N2 are $T_1$ 49bAG_s2 plants with a T-DNA insertion, while N3 and N4 are $T_1$ 49bAG_s2 plants without a T-DNA insertion

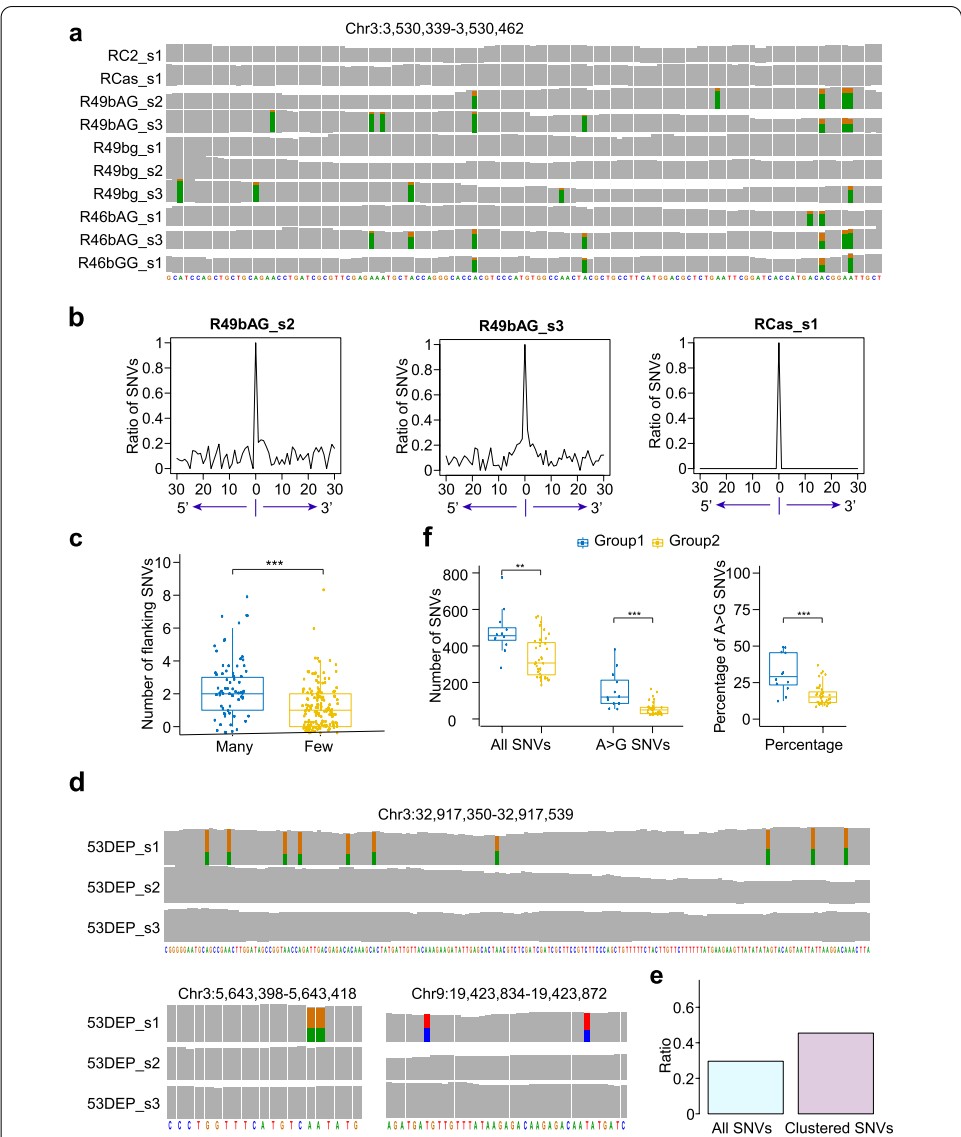

**Fig. 5** ABE-induced clustered RNA and DNA A>G SNVs. **a** An IGV genome browser view showing representative loci with clustered A>G SNVs in transcriptomes. **b** Ratios of A>G mutations were calculated in flanking 5′ and 3′ 30-bp regions centered at A>G RNA SNV loci. Lines R49bAG_s2 and R49bAG_s3 with RNA mutations and line RCas_s1 with SpCas9 only are shown. **c** Boxplot showing number of A>G SNVs in the flanking 5′ and 3′ 30-bp regions separately for RNA SNVs in many (3–8) or few (1–2) plants. **d** IGV genome browser views showing representative SNV loci with flanking A>G SNVs in whole-genome sequencing. **e** Ratios of clustered SNVs located in genic regions. **f** Plants with ABEs were classified into two groups: group 1 with clustered SNVs and group 2 without clustered SNVs. Number of SNVs and A>G SNVs, and percentage of A>G SNVs are shown separately for plants in group 1 and plants in group 2. (**) denotes *p*-value < 0.01, and (***) denotes *p*-value < 0.001 (one-tailed Wilcoxon test). In IGV genome browser views, the grey bar represents a sequenced nucleotide that is the same as the reference genome, while bars in other colors represent sequenced nucleotides that are partially or totally different from the reference genome: red represents nucleotide A, green represents nucleotide T, orange represents nucleotide G, and blue represents nucleotide C. The height of each color bar represents the relative composition of each nucleotide

to SNVs with flanking SNVs as clustered SNVs. By contrast, no flanking A>G editing occurred in plants lacking RNA off-target SNVs or in plants harboring SpCas9 nuclease (Fig. 5b and Additional file 2: Fig. S15 and S16). Of these A>G off-target RNA editing

events, there were SNVs with a high number of flanking A>G mutations and high occurrence in many transcriptomes, and there were also SNVs with a low number of flanking A>G mutations and occurrence in a few transcriptomes (Fig. 5a and Additional file 2: Fig. S17).

We performed similar studies on DNA off-target SNVs but did not observe general patterns of flanking A>G editing. However, we did identify 25 loci with more than one A>G SNV from 12 plants (Additional file 1: Table S10); some loci contained 5–10 A>G SNVs, and others contained 2–3 A>G SNVs (Fig. 5d and Additional file 2: Fig. S18). Overall, 45% of these SNVs were located in the genic region, which is higher than the 30% observed for all A>G SNVs in the genic region, consistent with the tendency of off-target A>G SNVs to occur in the genic region (Fig. 5e). We classified these 12 plants into group 1, and the remaining 36 plants carrying ABEs into group 2. The number of SNVs and A>G SNVs and the percentage of A>G SNVs were significantly higher for plants in group 1 compared to plants in group 2 (Fig. 5f).

## Discussion

The targeting specificity of CRISPR tools in applications remains a considerable concern. It is well known that Cas nucleases mediate highly specific genome editing with rare off-target mutations in plants [29, 30], and high-activity CBEs cause genome-wide off-target mutations in rice and mouse [14, 31, 32]. ABE8s and ABE9s have been developed by several groups to overcome the limitation of ABE7s [15–17]. Their robust editing efficiency raised another question: How is the specificity of those high-activity ABEs engineered with TadA8e and TadA9 deaminases? Compared to mouse and human genomes (each ~3 Gb), the rice genome (~0.4 Gb) is small, making WGS of individuals more feasible. In addition, rice is self-pollinating, circumventing the challenges of population heterogeneity of human cells, and lacks innate A-to-I RNA editing, facilitating analyses of ABE-induced RNA editing. Therefore, we performed a comprehensive evaluation of ABE8- and ABE9-induced genetic mutations through WGS and transcriptome sequencing in rice.

Cas proteins and TadA variants play different roles in ABE-induced DNA off-target mutations: ABEs harboring SpCas9n-NG, an engineered SpCas9 protein recognizing a flexible protospacer adjacent motif (PAM) [33–38], result in a higher number of total SNVs; those harboring TadA9, a TadA variant with robust activity [16], lead to a higher number of specific A>G SNVs. Plants transformed with the ABE rBE46b (SpCas9n-TadA8e) did not have more SNVs or a higher percentage of A>G SNVs than plants subjected to *Agrobacterium* infection, suggesting that selection of SpCas9n and TadA8e eliminates most sgRNA-independent DNA mutations induced by ABEs. Given that no sgRNA-dependent off-target mutations were observed, we conclude that optimization of sgRNA design is an efficient way of eliminating sgRNA-dependent off-target mutations.

Using deeply sequenced genomes and transcriptomes, we systematically studied ABE-induced RNA mutations. ABEs induce RNA A>G mutations in one-third of plants with high ABE expression but do not induce mutations in two thirds of plants with low ABE expression. When ABEs segregated out, RNA mutations diminished. In addition, T-DNA integration analysis suggested that stable ABEs induce more off-target SNVs than those whose T-DNA has not been integrated into the genome. Together, these data

highlight the importance of controlling the expression of ABEs in future applications, such as using inducible or photoactivatable transcription systems, ribonucleoprotein-based delivery in clinic gene therapy [39, 40], and transgene-free gene-edited plants in crop breeding.

Without the noise from A-to-I mutations mediated by ADAR proteins, we were able to obtain a clean set of ABE-induced RNA mutations and discovered that ABEs induced clustered A>G mutations, which provided useful information for defining and characterizing true ABE RNA targets. Furthermore, given the existence of common and unique mutations in plants regenerated from the same callus, we provide robust experimental evidence that plants with different on-target editing could be derived from the same T-DNA insertion event with a shared set of off-target SNVs. Therefore, we highly recommend using two independent transgenic lines from separated calli (with two different T-DNA insertion sites and two sets of non-overlapping SNVs) in gene function studies.

## Conclusions

The properties of the small genome, self-pollination, and the absence of ADAR proteins make rice a model organism to employ large-scale sequencing approaches to evaluate ABEs' off-target activity. The pioneering comprehensive analysis of ABE-induced DNA and RNA mutations using whole-genome and transcriptome sequencing in rice sheds light on defining and characterizing ABEs' specificity. The discovery that Cas proteins, TadA variants, transient expression, and the expression level of ABEs contribute to ABEs' specificity in rice points out alternative ways improving ABEs' specificity including combinatorial optimization of Cas/deaminase (SpCas9n-TadA8e) and temporal control of ABEs' expression besides the traditional protein engineering of deaminases.

## Materials and methods

### Plasmid construction

In this study, five rice (*Oryza sativa*) genomic loci (*OsACC*, *OsGS1*, *OsMPK13*, *OsGSK3*, and *OsGSK4*) and four rice genomic loci (*OsACC*, *OsGS1*, *OsMPK13*, and *OsTms9*) were targeted by rBE46b and rBE49b, respectively. Three genes (*OsSERK2*, *OsDEP2*, and *OsGSK4*) were targeted by both rBE50 and rBE53. Plant IDs and their corresponding information are described in Additional file 1: Table S1. The rBE46b, rBE49b, rBE50, and rBE53 expression plasmids were constructed as previously reported [16]. The empty entry vector without any spacer was cloned into pUbi:rBE46b, pUbi:rBE49b, pUbi:rBE50, and pUbi:rBE53 using Gateway technology to yield ABEs without sgRNAs (Additional file 1: Table S1).

### *Agrobacterium*-mediated rice transformation and plant growth

The genome editing constructs were individually introduced into the *Agrobacterium tumefaciens* strain EHA105 via the freeze-thaw transformation method, and 2-week-old calli derived from immature seeds of the Geng rice variety Kitaake were infected by each *Agrobacterium* strain. After 4 weeks of culture on MSD medium supplemented with 50 mg/L hygromycin (Roche, Germany), the resistant callus lines were transferred onto RM plates to generate transgenic rice seedlings. All information on target gene mutations of each seedling examined in this study is given in Additional file 1: Table S1.

To eliminate background mutations, 10 individual Kitaake plants grown from seeds were used directly. Seedlings were regenerated from rice calli without *Agrobacterium* infection (namely C1) and regenerated from calli co-cultured with the empty EH105 strain (namely C2). Also, seedlings were regenerated from calli infected with EH105 strains harboring SpCas9 only (namely Cas). All rice materials were grown in the greenhouse under a 16-h-light/8-h-dark photoperiod, 28/25 °C temperature cycle, and 75% humidity.

### DNA and RNA extractions

Genomic DNA of 4-week-old rice plants was extracted using the CTAB method (Li et al., 2016). Approximately 200 mg of fresh rice leaves was collected in a 2-ml centrifuge tube containing disposable metal balls. After being quickly frozen in liquid nitrogen, samples were ground to a fine powder using a tissue grinding apparatus (Jingxin, China). Following chloroform extraction, isopropanol precipitation, and 70% EtOH washing, genomic DNAs were eluted with 50 μL of double-distilled water supplemented with 1 μL of 10 U/μL RNase I (Thermo Fisher Scientific, USA) and stored at −80 °C for later experiments.

RNA was extracted with TRIzol reagent (Takara, Japan) according to the manufacturer's instructions. Briefly, 100 mg of fresh rice leaves was sampled, quickly frozen in liquid nitrogen, and ground to a powder with a tissue grinding apparatus. Then, 1 ml of TRIzol reagent was added to the sample followed by chloroform and isopropanol treatment. Finally, RNA pellets were dissolved in 50 μL of RNase-free water (0.1% DEPC-treated) and stored at −80 °C for later experiments.

### Detection and validation of on-target and off-target mutations

The on-target genomic regions were amplified using Phanta Max Super-Fidelity DNA Polymerase (Vazyme, China) and locus-specific primers (Additional file 1: Table S1, Table S11, Table S12) with genomic DNAs and cDNAs used as the template. PCR amplicons were subjected to Sanger sequencing, and Bioedit software was used for sequence data analysis.

### Whole-genome analysis of genetic mutations

RNA-free genomic DNAs (0.2 μg) from each sample were used to construct the DNA libraries using a NEBNext Ultra DNA Library Prep Kit for Illumina (NEB, USA) following the manufacturer's instructions. DNA libraries were sequenced on the Illumina platform in the 150-nt paired-end mode with an average coverage depth of 40× (Additional file 1: Table S2).

The clean reads were mapped to the Kitaake genome V3 from Phytozome (https://data.jgi.doe.gov/refine-download/phytozome) via BWA [41] and sorted using samtools (v1.9) [42]. The Genome Analysis Toolkit (GATK v4.2) was used to mark duplicated reads and recalibrate base qualities [25]. To identify high-quality genetic changes at the genomic scale, we applied three independent germline variant-calling methods: GATK, LoFreq [23], and Strelka2 [23]. We documented SNVs identified by all three methods and indels identified by GATK and Strelka. All genetic changes identified by the three methods in the 10 Kitaake plants were combined and used as background mutations. Sanger sequencing has been performed to validate the overlapping set of SNVs called

by the three methods (Additional file 2: Fig. S19). The genetic mutation ratios were calculated using an in-house R program and 'AC' value from GATK's results. Both background mutations and homozygous mutations were removed from the SNVs as well as indels. The IGV browser was used to demonstrate sgRNA-directed on-target mutations [43]. Then, the on-target mutations were removed for off-target analysis. sgRNA-dependent off-target mutations were discovered using Crisflash [26], and the genetic on-target mutations were assessed using the IGV browser. A gene annotation file (OsativaKitaake_499_v3.1.gene_exons.gtf) from the Phytozome website was used to define different genomic regions, such as gene regions, exon regions, and intergenic regions. The ggpubr, ggbio, and VennDiagram R libraries were used to draw the graphs.

### Analysis of T-DNA insertion sites and ABE transcripts

The clean reads were mapped to T-DNA sequences using BWA and sorted using samtools. The T-DNA insertion sites were located through T-LOC (Li et al. in preparation). The coverage of T-DNAs between the left border (LB) and right border (RB) was assessed using the R library ShortRead. The expression of ABEs was quantified as the average raw read number of Cas proteins and TadA variants normalized by the total read number in millions. Since we used T0 plants, the copy number of T-DNA integration was calculated as the relative T-DNA coverage versus half coverage of the rice genome.

### Analysis of ABE-induced RNA mutations

DNA-free RNAs (0.2 μg) were used to construct the RNA-seq libraries using a NEB Next Ultra RNA Library Prep Kit for Illumina (NEB, USA) following the manufacturer's instructions. RNA-seq libraries were sequenced on the Illumina platform in the 150-nt paired-end mode (Additional file 1: Table S8).

The clean reads were mapped to the Kitaake V3 genome and annotation from Phytozome via STAR aligner with a maximum of eight mismatches per paired-end read [44]. GATK was used to mark duplicate reads and split reads that contained Ns in their cigar string and to recalibrate base qualities. SNVs were called by GATK, LoFreq, and Strelka2 for each transcriptome dataset and corresponding genome dataset. The SNVs identified by three methods in the transcriptome data but not in the genome data were kept for later analysis. Sanger sequencing has been performed to validate the overlapping set of SNVs called by the three methods (Additional file 2: Fig. S20). All the genetic changes identified by the three methods in three *Agrobacterium*-infected plants were combined and used as background mutations and were removed from the SNVs identified in plants transformed with SpCas9, rBE46b, and rBE49b. The A>G mutation ratios of off-target RNA loci were calculated through in-house Python programs. The 30- and 3-bp flanking sequences of the off-target RNA SNVs were extracted from the Kitaake reference genome and subjected to motif prediction using WebLogo3 (http://weblogo. threeplusone.com/) [45].

### Calculation of flanking A>G mutations in genome and transcriptome data

We combined all A>G off-target SNVs obtained from plants with RNA off-target activities. For each A>G SNV, we calculated the number of reads with nucleotide A, T, G, and C separately in the 5′ and 3′ 30-bp region with a read coverage larger than 10. The

genetic change ratio was calculated as the number of Gs divided by the total number of As and Gs if the reference is A. The genetic change ratio was calculated as the number of Cs divided by the total number of Cs and Ts if the reference is T. Positions with an A>G mutation ratio of higher than 0.05 were used as the numerator, while positions of A/T with a read coverage larger than 10 were used as the denominator. Similarly, we combined all A>G off-target SNVs obtained from plants through WGS and calculated the percentage of A>G mutations at the 5′ and 3′ 30-bp flanking regions.

### Parameters of boxplots used in this study

The horizontal line in the box represents the median value, and the bottom and top of the box are the lower (Q1) and upper quartiles (Q3), respectively. The upper whisker is $\min(\max(x), Q3 + 1.5 \times IQR)$, and the lower whisker is $\max(\min(x), Q1 - 1.5 \times IQR)$. IQR (interquartile range) $= Q3 - Q1$. Black dots located outsides the whiskers are outliers.

### Supplementary Information

---

**Additional file 1:** Supplementary tables. **Table S1**, Summary of plants with ABEs. **Table S2**, Mapping statistics of whole-genome sequencing. **Table S3**, Summary of sgRNA-dependent on-target and off-target loci. **Table S4**, Summary of all the homozygous SNVs. **Table S5**, Summary of all the homozygous indels. **Table S6**, Summary of genomic SNVs detected through WGS. **Table S7**, Summary of the overlapping SNVs between each of the plants with whole-genome sequencing. **Table S8**, Mapping statistics of whole-transcriptome sequencing. **Table S9**, Summary of all the transcriptomic SNVs. **Table S10**, Summary of clustered A>G DNA SNVs. **Table S11**, Primers used to verify DNA SNVs by Sanger sequencing. **Table S12**, Primers used to verify RNA SNVs by Sanger sequencing.

**Additional file 2:** Supplementary figures. **Fig. S1**. Sanger sequencing chromatograms of on-target mutations in plants harboring rBE46b and rBE49b. **Fig. S2**. Sanger sequencing chromatograms of on-target mutations in plants harboring rBE50 and rBE53. **Fig. S3**. IGV browser views showing the on-target mutations for 36 plants harboring ABEs. **Fig. S4**. Analysis of SNVs and indels identified by whole-genome sequencing. **Fig. S5**. Analysis of the remaining background homozygous DNA mutations. **Fig. S6**. Characterization of ABE-induced genomic mutations. **Fig. S7**. Distribution of six types of SNVs. **Fig. S8**. Distribution of SNVs at given regions of the genome. **Fig. S9**. Chromosomal distribution of SNVs. **Fig. S10**. On-target and off-target mutations in plants from the same calli. **Fig. S11**. Off-target SNVs in plants with incomplete T-DNA insertions. **Fig. S12**. Distribution of SNVs with different copy numbers of T-DNA insertions. **Fig. S13**. Transcriptome-wide distribution of ABE-induced off-target mutations. **Fig. S14**. Heatmap demonstrating A>G mutations in transcriptomes with more than 5 A>G SNVs detected. **Fig. S15**. The 5′ and 3′ flanking A>G mutations in transcriptomes with ABEs containing A>G RNA SNVs and in transcriptomes with SpCas9 only lacking A>G RNA SNVs. **Fig. S16**. The 5′ and 3′ flanking A>G mutations in transcriptomes with ABEs but without A>G RNA SNVs. **Fig. S17**. IGV genome browser views showing the off-target RNA mutations. **Fig. S18**. IGV genome browser views showing A>G mutations with flanking A>G SNVs in genome sequencing data. **Fig. S19**. Sanger sequencing chromatograms of off-target A>G DNA mutations. **Fig. S20**. Sanger sequencing chromatograms of off-target A>G RNA mutations.

**Additional file 3:** Review history

---

### Acknowledgements

We thank Sujie Zhang and Yongjie Kuang for assistance with RNA manipulation.

### Peer review information

### Authors' contributions

S.L., W.S., X.Z., and H.Z. designed and guided the research. S.L. performed bioinformatic analysis and L.L. performed experiments. S.L. and H.Z. wrote the manuscript. All authors read and approved the final manuscript.

### Funding

This work was supported by grants from the National Natural Science Foundation of China (31871948) and the Central Public-interest Scientific Institution Basal Research Fund (Y2020PT26) to H.Z.

### Availability of data and materials

All data in the study has been included in the manuscript and additional files. All sequencing genome and transcriptome data have been deposited in the NCBI database with the accession number GSE185497 [46].

## Declarations

### Ethics approval and consent to participate
Not applicable.

### Competing interests
The authors declare that they have no competing financial interests.

### Author details
[1]State Key Laboratory for Biology of Plant Diseases and Insect Pests, Institute of Plant Protection, Chinese Academy of Agricultural Sciences, Beijing 100193, China. [2]Scientific Observing and Experimental Station of Crop Pests in Guilin, Ministry of Agriculture and Rural Affairs, Guilin 541399, China. [3]Department of Plant Pathology, China Agricultural University, Beijing 100193, China. [4]State Key Laboratory of Rice Biology, Institute of Biotechnology, Zhejiang University, Zhejiang, Hangzhou, China.

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

## 

