## [**Additional file 3:** Review history · Genome Biology]

Review History

First round of review

Reviewer 1

Are you able to assess all statistics in the manuscript, including the appropriateness of statistical tests used? Yes, and I have assessed the statistics in my report.

Comments to author:

ABEs show robust base editing activity but raise concerns about off-target effects. The authors performed a comprehensive evaluation of ABE8e- and ABE9-induced DNA and RNA mutations in *Oryza sativa*. The WGS analysis and transcriptome analysis showed that Cas proteins (SpCas9n and SpCas9n-NG), TadA variants (ABE8e and ABE9), temporal expression of ABEs, and expression levels of ABEs contribute to ABE specificity in rice. This work includes interesting results and experimental data supports the conclusion. However, I think that this manuscript has many points that should be revised.

Major comments:

1. For the key SNVs obtained from WGS analysis and transcriptome data analysis, has the author verified the key SNVs? For example, the comparison between TadA8e and TadA9, SpCas9n and SpCas9n-NG obtained by WGS analysis; the comparison between RNA editing and without RNA editing obtained by transcriptome analysis, etc.
2. WGS analysis indicates the randomness of off-target DNA mutations induced by tissue culture, *Agrobacterium* infection, and ABEs. Whether there is similarity between off target and on target gRNA sequences, whether off target editing is caused by poor specificity of off target sequence selection, and whether off target gRNA mutations can be predicted by on target gRNA mutations.
3. Transcriptome analysis indicated that ABEs might have preferred RNA editing sequence content, and identified a conserved YAN-enriched motif at ABE-edited RNA loci. However, compared to RNA mutations, DNA mutations are completely random and irregular. Whether this is caused by excessive background values detected at the DNA level is unknown. The authors should further explain this phenomenon.

Minor comments:

4. Line 96, "except 46bM and 49bM" should be revised to "except 46bM and 49bTm". Please recheck this.
5. Line 210, "Additional file 1: Figure S11b" should be revised to "Figure S10b". Please recheck this.
6. Line 249, The author only shows the correlation of the ratio of ABE-induced RNA off-target SNVs in transcriptome R49AG_s2 and R49AG_s3, the correlation between other RNA mutant plants should be supplemented at the same time.
7. Line 266-267: "between the plants with RNA mutations and those without RNA mutations (Additional 267 file 1: Figure S12f)". Figure S12f is missing from the manuscript. Please revise it.

Reviewer 2

Are you able to assess all statistics in the manuscript, including the appropriateness of statistical tests used? Yes, and I have assessed the statistics in my report.

Comments to author:

The manuscript submitted by Li et al. (GBIO-D-21-01717) report on deeply analysis of adenine base editors, ABE8e and ABE9, off-target DNA and RNA mutations in rice with WGS strategy. The authors found that both Cas and ABEs contributes to the number of A-to-G off-target mutations. Different situations of T-DNA integration have been examined. Besides, they also analyzed transcriptome-wide ABE induced RNA mutations. This study is very interesting and will draw great attentions to plant genome editing community. So, in my opinion, I think it is proper to be published on Genome Biology after minor revision. Here are some comments for revision:

1. For Agrobacterium-mediated plant transformation, T-DNA could be integrated into plant genome in one or multiple copies. The analysis between the number of ABEs copies and its effects on ABE-induced mutations should be provided.
2. To provide enough background information and make comprehensive analysis, some key references for plant genome editing mutation analysis (Feng et al., Proc Natl Acad Sci U S A 2014; Tang et al., Genome Biol 2018, 19(1):84; Li et al., Plant Biotechnol J 2018; Ren et al., Plant Biotechnol J 2021) and SpyCas9/SpyCas9 PAM relaxed variants based base editing (Hua et al., Mol Plant 2019; Zhong et al., Mol Plant 2019; Qin et al., Nat Plants 2020; Li et al., Nat Biotechnol 2020; Ren et al., Nat Plants 2021) could be included in this manuscript and give appropriate discussions.
3. The panel b in figure 1 was a little bit confusing. The improvements should be made to better describe the experimental design ideas. Also, "Tissue culture" and "Agro transformation" control samples could be described more clearly in "Materials and Methods" part.
4. In L192-200 and Fig 3A-B, 49bM_s2 has the same T-DNA insertion with 49bM_s3. However, in Table S1, it was labelled that 49bTm_s2 and 49bTm_s3 has the T-DNA insertion. Please correct this inconsistency.
5. In table S1, an empty row was placed above 50bg_s1, please correct this format.
6. Four 49bAG_s2 T1 transcriptome data were missed from NCBI GEO submission.
7. In table S6, table S9 and table S10, please use A>G or A-to-G rather than A_G to clearly label SNV mutation type.

Dec 31st, 2021
Editorial Team
Genome Biology

Dear Editors,

Thank you very much for considering our manuscript “**A large-scale genome and transcriptome sequencing analysis reveals the mutation landscapes induced by high-activity adenine base editors in plants**” for publication in *Genome Biology*. We have carefully considered the comments and suggestions and revised our manuscript accordingly. We have fully addressed the two reviewers’ concerns and made GEO submission public. We hope that the revisions are accepted and that our responses adequately address the comments.

Thank you very much for your time and consideration of this manuscript.
We are looking forward to hearing from you.

Best regards

Dr. Huanbin Zhou

Dr. Shaofang li

Institute of Plant Protection, Chinese Academy of Agricultural Sciences

Beijing, China, 100193

Email: shaofangli2021@hotmail.com or hbzhou@ippcaas.cn

Point-by-point Responses

(Responses to each comment are written below in red typeface.)

First, we fully appreciate the reviewer’s constructive comments and suggestions that have allowed us to make our revised manuscript much better than the original.

Reviewer #1: ABEs show robust base editing activity but raise concerns about off-target effects. The authors performed a comprehensive evaluation of ABE8e- and ABE9-induced DNA and RNA mutations in *Oryza sativa*. The WGS analysis and transcriptome analysis showed that Cas proteins (SpCas9n and SpCas9n-NG), TadA variants (ABE8e and ABE9), temporal expression of ABEs, and expression levels of ABEs contribute to ABE specificity in rice. This work includes interesting results and experimental data supports the conclusion. However, I think that this manuscript has many points that should be revised.

Major comments:

1. For the key SNVs obtained from WGS analysis and transcriptome data analysis, has the author verified the key SNVs? For example, the comparison between TadA8e and TadA9, SpCas9n and SpCas9n-NG obtained by WGS analysis; the comparison between RNA editing and without RNA editing obtained by transcriptome analysis, etc.

We thank the reviewer for this concern. We have performed sanger sequencing validation of A>G SNVs obtained through whole-genome and transcriptome (L443-444, L477-478, Additional file 1, Figure S19, Figure S20 and additional file2: Table S11, Table S12). In addition, we have applied six different criteria of obtaining SNVs in addition to the overlapping set of GATK, Strelka2 and Lofreq and have the same conclusion (FigureRevision_1).

Figure legend: A-C, Seven sets of SNVs from each sequenced genome were obtained using seven criteria. Sets 1-3 were obtained using GATK, Strelka2 or Lofreq separately. Sets 4-6 were obtained as the overlapping set using GATK and Strelka2, GATK and Lofreq, Strelka2 and Lofreq. Sets 7 were obtained as the overlapping using GATK, Strelka2 and Lofreq (Criteria suggested by Jin et al 2020 Science and used in the manuscript). We carried out wilcox.test using these seven sets of SNVs separately. (A-B), Seven p -values were obtained on the number of SNVs and percentage of A>G SNVs for each of the following comparisons: rBE46b vs rBE49b (p -values in p1), rBE50 vs rBE53 (p -values in p2), rBE46b vs rBE50 (p -values in p3), rBE49b vs rBE53 (p -values in p4). When $-\log_{10} p$ -value is lower than 1, the comparison is not significant, when $-\log_{10} p$ -value is higher than 1, the comparison is significant. From FigureRevision_1 A, we can see that the total number of SNVs is not higher in rBE49b when compared with rBE46b and that the total number of SNVs is not higher in rBE50 when compared with rBE53. We can see that the total number of SNVs is higher in rBE50 when compared with rBE46b and that the total number of SNVs is higher in rBE53 when compared with rBE49b. These indicates that SpCas9n-NG leads to a higher total number of SNVs. Similarly, we can see TadA9 leads to a higher percentage of A>G SNVs from FigureRevision_1 B. In FigureRevision_1 C, Seven p -values were obtained on the number of SNVs (pA), the number of A>G SNVs (pB) and percentage of A>G SNVs (pC) between plants with A>G RNA editing and plants without A>G RNA editing. The $-\log_{10} p$ -values that are lower than 1 indicate there are no significant differences.

2. WGS analysis indicates the randomness of off-target DNA mutations induced by tissue culture, Agrobacterium infection, and ABEs. Whether there is similarity between off target and on target gRNA sequences, whether off target editing is caused by poor specificity of off target sequence selection, and whether off target gRNA mutations can be predicted by on target gRNA mutations.

We thank the reviewer for their concern. The randomness of off-target DNA mutation and the sequence specificity are two contradictory facts. In the manuscript, we showed that the gRNA-dependent off-target mutations do not occur at 33 off-targeted loci predicted by Criflash with 3 maximal mismatches. To further address this question, we predicted 33,558 (12991 from rBE46b and rBE49b and 20567 from rBE50 and rBE53) gRNA-dependent off-target loci with 6 nt maximal mismatches and only detected 5 SNVs at these loci. In addition, we manually located 366,136 loci through searching the 'seed' region consisting of 12 bases closer to the 3' end of the spacer allowed 2 nt mismatches and only detected 35 SNVs at these loci. All of these (Manuscript L114-122) demonstrate that off-target mutations can't be predicted by gRNA sequences with mismatches.

3. Transcriptome analysis indicated that ABEs might have preferred RNA editing sequence content, and identified a conserved YAN-enriched motif at ABE-edited RNA loci. However, compared to RNA mutations, DNA mutations are completely random and irregular. Whether this is caused by excessive background values detected at the DNA level is unknown. The authors should further explain this phenomenon.

This question is related to question 2. Similarly, since DNA mutations are completely random and irregular, we shouldn't detect any motif at DNA mutation loci. However, I agree with the reviewer on the fact that off-target DNA SNVs contains excessive background values mutations that are not A>G SNVs. Here are some explanations. First, TadAs are tRNA-specific adenosine deaminase and it is acceptable that TadAs has some leaky activity on mRNA with a low activity. The low number of A>G SNVs and low ratio of A>G SNVs suggests that ABEs induce A>G RNA mutation at a very low ratio. Second, we didn't discover any motif on 31 ABEs-Edited-As and 53 ABEs-UnEdited-A from 14 ABEs sgRNAs on-target loci. Third, no motifs were found on cluster A>G SNVs as well as no shared off-target SNVs were found in plants with same gRNAs. Hope this could address the reviewer's concern.

Minor comments:

4. Line 96, "except 46bM and 49bM" should be revised to "except 46bM and 49bTm". Please recheck this.

We appreciate the careful reading of the reviewer. There was a typo in Table S1 and 49bM_s2 has the same T-DNA insertion with 49bM_s3. I have corrected this typo in Table S1.

5. Line 210, "Additional file 1: Figure S11b" should be revised to "Figure S10b". Please recheck this.

We appreciate the careful reading of the reviewer. This typo has been revised (L216).

6. Line 249, The author only shows the correlation of the ratio of ABE-induced RNA off-target SNVs in transcriptome R49AG_s2 and R49AG_s3, the correlation between other RNA mutant plants should be supplemented at the same time.

Thank you for your attention. As suggested, we demonstrated the commonly shared A>G SNVs in eight transcriptomes with RNA editing in a heatmap (Additional file 1: Figure S14 and L263-264).

7. Line266-267: "between the plants with RNA mutations and those without RNA mutations (Additional 267 file 1: Figure S12f)". Figure S12f is missing from the manuscript. Please revise it.

Thank you for your attention. "Figure S12f" has been changed to "Figure S13e" in revised manuscript (L280).

Reviewer #2: The manuscript submitted by Li et al. (GBIO-D-21-01717) report on deeply analysis of adenine base editors, ABE8e and ABE9, off-target DNA and RNA mutations in rice with WGS strategy. The authors found that both Cas and ABEs contributes to the number of A-to-G off-target mutations. Different situations of T-DNA integration have been examined. Besides, they also analyzed transcriptome-wide ABE induced RNA mutations. This study is very interesting and will draw great attentions to plant genome editing community. So, in my opinion, I think it is proper to be published on Genome Biology after minor revision. Here are some comments for revision:

1. For Agrobacterium-mediated plant transformation, T-DNA could be integrated into plant genome in one or multiple copies. The analysis between the number of ABEs copies and its effects on ABE-induced mutations should be provided.

That's a good question. As suggested, we calculated the correlation between the copy number of T-DNA integrations and ABE-induced DNA mutations (L236-241, L462-464 and Additional file 1: Figure S12). It turns out that the copy number of T-DNA integrations does not impact ABE-induced DNA mutations.

2. To provide enough background information and make comprehensive analysis, some

key references for plant genome editing mutation analysis (Feng et al., Proc Natl Acad Sci U S A 2014; Tang et al., Genome Biol 2018, 19(1):84; Li et al., Plant Biotechnol J 2018; Ren et al., Plant Biotechnol J 2021) and SpyCas9/SpyCas9 PAM relaxed variants based base editing (Hua et al., Mol Plant 2019; Zhong et al., Mol Plant 2019; Qin et al., Nat Plants 2020; Li et al., Nat Biotechnol 2020; Ren et al., Nat Plants 2021) could be included in this manuscript and give appropriate discussions.

Thanks for the suggestion, all references have been included and discussed in the revised manuscript (L64-65, L320-323, L335).

3. The panel b in figure 1 was a little bit confusing. The improvements should be made to better describe the experimental design ideas. Also, "Tissue culture" and "Agro transformation" control samples could be described more clearly in "Materials and Methods" part.

We thank the reviewer for their concern. Fig. 1b has been revised and more details have been given to controls in Materials and Methods (L398-402).

4. In L192-200 and Fig 3A-B, 49bM_s2 has the same T-DNA insertion with 49bM_s3. However, in Table S1, it was labelled that 49bTm_s2 and 49bTm_s3 has the T-DNA insertion. Please correct this inconsistency.

We appreciate the careful reading of the reviewer. We have corrected this as stated in the response to opinion 4 from Reviewer #1.

5. In table S1, an empty row was placed above 50bg_s1, please correct this format.

As suggested, the empty row was removed.

6. Four 49bAG_s2 T1 transcriptome data were missed from NCBI GEO submission.

Thanks for the reviewer's opinion. We have submitted these transcriptomes and made GEO submission public.

7. In table S6, table S9 and table S10, please use A>G or A-to-G rather than A_G to clearly label SNV mutation type.

Thanks for the opinion. As suggested, all of “_”s have been changed to “>” in table S6, S9 and S10.

Second round of review

Reviewer 1

In the revised version of the manuscript, the authors had added additional data and answered the questions that the Reviewers raised. In my opinion, the manuscript can be published in the Genome Biology.